# Quality Assessment of an Antimicrobial Resistance Surveillance System in a Province of Nepal

**DOI:** 10.3390/tropicalmed6020060

**Published:** 2021-04-23

**Authors:** Jyoti Acharya, Maria Zolfo, Wendemagegn Enbiale, Khine Wut Yee Kyaw, Meika Bhattachan, Nisha Rijal, Anjana Shrestha, Basudha Shrestha, Surendra Kumar Madhup, Bijendra Raj Raghubanshi, Hari Prasad Kattel, Piyush Rajbhandari, Parmananda Bhandari, Subhash Thakur, Saroj Sharma, Dipendra Raman Singh, Runa Jha

**Affiliations:** 1National Public Health Laboratory, Kathmandu 44600, Nepal; nisharijal1@gmail.com (N.R.); anjananshup@gmail.com (A.S.); runa75jha@gmail.com (R.J.); 2Institute of Tropical Medicine, 2000 Antwerp, Belgium; mzolfo@itg.be; 3Department of Dermatology and Venereology, BahirDar University, 1996 Bahir Dar, Ethiopia; wendaab@gmail.com; 4Amsterdam UMC, Academic Medical Centre, Department of Dermatology, Amsterdam Institute for Infection and Immunity (AI&I), University of Amsterdam, 7057 Amsterdam, The Netherlands; 5International Union against Tuberculosis and Lung Disease, Paris, France and International Union against Tuberculosis and Lung Disease, Mandalay 11061, Myanmar; dr.khinewutyeekyaw2015@gmail.com; 6World Health Organization, Health Emergencies Unit, Kathmandu 44700, Nepal; meikabhattachan@gmail.com; 7Kathmandu Model Hospital, Kathmandu 44600, Nepal; basudha111@gmail.com; 8Dhulikhel Hospital, Dhulikhel 45200, Nepal; sur2036@hotmail.com; 9KIST Medical College Teaching Hospital, Lalitpur 44700, Nepal; raghu2_47@yahoo.co.in; 10Tribhuwan University Teaching Hospital, Kathmandu 44600, Nepal; kattelhari@hotmail.com; 11Patan Hospital, Patan Academy of Health Sciences, Lalitpur 44700, Nepal; piyushrajbhandari@pahs.edu.np; 12Sukraraj Tropical and Infectious Disease Hospital, Kathmandu 44600, Nepal; parmananda_bhandari@yahoo.com; 13Paropakar Maternity and Women’s Hospital, Kathmandu 44600, Nepal; subasmt5@gmail.com; 14Kanti Children’s Hospital, Kathmandu 44600, Nepal; sarose.sharma30@yahoo.com; 15Quality Standard and Regulation Division, Ministry of Health and Population, Kathmandu 44600, Nepal; dipendra2028@gmail.com

**Keywords:** Global Antimicrobial Resistance Surveillance System (GLASS), reference laboratory, NPHL, SORT IT, operational research

## Abstract

Antimicrobial resistance (AMR) is a global problem, and Nepal is no exception. Countries are expected to report annually to the World Health Organization on their AMR surveillance progress through a Global Antimicrobial Resistance Surveillance System, in which Nepal enrolled in 2017. We assessed the quality of AMR surveillance data during 2019–2020 at nine surveillance sites in Province 3 of Nepal for completeness, consistency, and timeliness and examined barriers for non-reporting sites. Here, we present the results of this cross-sectional descriptive study of secondary AMR data from five reporting sites and barriers identified through a structured questionnaire completed by representatives at the five reporting and four non-reporting sites. Among the 1584 records from the reporting sites assessed for consistency and completeness, 77–92% were consistent and 88–100% were complete, with inter-site variation. Data from two sites were received by the 15th day of the following month, whereas receipt was delayed by a mean of 175 days at three other sites. All four non-reporting sites lacked dedicated data personnel, and two lacked computers. The AMR surveillance data collection process needs improvement in completeness, consistency, and timeliness. Non-reporting sites need support to meet the specific requirements for data compilation and sharing.

## 1. Introduction

The increasing prevalence of antimicrobial resistance (AMR) is a global public health threat, recognized by the World Health Organization (WHO) member states and the World Health Assembly as a priority area for global action [1].The AMR Global Action Plan (GAP), introduced by the WHO in 2015, has provided guidance to countries in tackling antibiotic resistance, with microbiological surveillance as a key action [2].

AMR surveillance is important for revealing patterns, trends, and outbreaks of resistant microorganisms at the national and international levels [3]. Data from this surveillance allow policymakers and health care providers to introduce evidence-based standards and regulations and to make appropriate decisions about antibiotic prescribing [4].

Countries are expected to report annually to the WHO regarding their progress on AMR data using the Global Antimicrobial Resistance Surveillance System (GLASS). The GLASS-AMR platform collects aggregated data on the frequency of antimicrobial resistance among high-priority pathogens that cause acute infections in humans, namely, *Escherichia coli, Klebsiella pneumoniae, Salmonella* spp., *Shigella* spp., *Acinetobacter* spp., *Streptococcus pneumoniae, Staphylococcus aureus*, and *Neisseria gonorrhoeae*. These pathogens are the most commonly identified in hospital-acquired and community-acquired infections, and treatment is becoming increasingly difficult because of high rates of AMR. Moreover, some of the selected bacteria included in the GLASS are present in non-human animals and the food supply. The data for the GLASS are collected at the national level and enable countries to share comparable and validated data worldwide to drive national, regional, and global actions in reducing the AMR burden [5].

Other than AMR data, the GLASS also is a repository of information about the status of national AMR surveillance systems, including the presence of a National Reference Laboratory (NRL), National Coordinating Centre (NCC), and National Action Plan (NAP) from enrolled countries. By April 2020, 91 countries and territories had enrolled in the GLASS, and 66 countries submit AMR data reports [5].

Laboratory-based surveillance systems require resources, facilities, training, and a central public health reference laboratory for quality assurance [6]. The Ministry of Health and Population in Nepal, with the participation of various multi-sectoral stakeholders, has drafted an NAP for AMR in line with the GAP and, in 2017, designated the National Public Health Laboratory (NPHL) as the NRL and NCC for Nepal.

The NPHL initiated a laboratory-based AMR surveillance about two decades ago, with nine sentinel sites [7], which has now expanded to 21 sites country-wide. Ten of these sites are situated in Province 3, one of the most populous provinces in Nepal [8]. The NPHL receives AMR data from the surveillance sites and is responsible for collating and combining the data before submission to the GLASS. The completeness, consistency, and frequency of data reporting vary among surveillance sites, with some sending the information monthly, others quarterly, and others only sporadically. The NPHL has adhered to the GLASS initiative and began to submit data to the platform in 2017 [9].

The aim of this study was to assess and verify the AMR surveillance laboratory data reported to the GLASS from five AMR “reporting” surveillance sites and to use a basic infrastructure assessment to identify barriers to submission for four “non-reporting” sites, all in Nepal’s Province 3. To our knowledge, this work represents the first such assessment of the quality of AMR data submitted through a surveillance system in Nepal.

## 2. Materials and Methods

### 2.1. Study Design

This is a cross-sectional descriptive study using secondary AMR data collection and a structured questionnaire.

### 2.2. Settings

#### 2.2.1. General Setting

Nepal is a landlocked country in Asia with a population of around 30 million. The country consists of seven self-governing provinces with 77 administrative districts and 753 local self-governing bodies. Province 3 (presently Bagmati Pradesh) represents 22% of the total country population. Moreover, it has the highest number of microbiological surveillance sites (10 out of the total 21 surveillance sites in the country), which are easily accessible and are larger sites with a relatively high volume of AMR data. For these reasons, we selected them for this study.

#### 2.2.2. Specific Setting

The NPHL is the national reference laboratory responsible for monitoring, supervising, capacity-strengthening, and providing an external quality assessment (EQA) of all microbiological surveillance sites in the country. The NPHL took on the responsibility of reporting to the GLASS in 2017, when only one site submitted complete data, which was not published in the GLASS. In 2018 and 2019, the reporting sites increased from 14 to 15 out of 21 that provided complete data to the GLASS platform [5]. The data are supposed to be received monthly from all the AMR surveillance sites and collated and compiled at the NPHL before submission to the GLASS.

The NPHL supports the surveillance sites through annual refresher trainings on “Identification of AMR surveillance pathogens and antimicrobial susceptibility testing”, data entry, deduplication and compilation, and on the use of the WHONET software, a freely downloadable database software developed for the management and analysis of antimicrobial susceptibility test results [10]. The surveillance site laboratories are trained on standard procedures of antimicrobial susceptibility testing following the annually updated Clinical and Laboratory Standards Institute (CLSI) criteria. The NPHL also regularly supervises the laboratories through periodic monitoring visits, although not all sites are necessarily visited each year. The NPHL further ensures the quality of laboratories by sending regular EQA scheme samples, followed by feedback on the laboratory performance.

Only nine of the ten sites located in Province 3 were considered for this study because one of the sites is an animal health site and thus not eligible for GLASS reporting. The selected surveillance sites are located at tertiary care or specialty hospitals with in-patient as well as out-patient departments in Province 3, except the NPHL which is the reference stand-alone laboratory. Five of the sites sending regular data to the NPHL were identified as “reporting” sites, whereas four that had failed to report data for three consecutive months were identified as “non-reporting” sites.

### 2.3. Study Population and Period

We reviewed detailed laboratory records of positive bacterial culture of AMR pathogens sent to the NPHL from January to June 2019 from the five “reporting” surveillance sites. The five “reporting” sites had variable numbers of specimens tested varying from 6000 to 36,000 with most being blood and urine specimens. The positive bacterial culture numbers also varied from site to site, so a random subset calculated from previous year’s records was selected for verification. The detailed laboratory records (range 171–428 per site, totaling to 1584 records) from each site, calculated on the basis of 2018 submitted GLASS data (total 89,553 records) through online OpenEpi software, were selected for verification. In addition, one microbiology laboratory staff involved in data compilation from each of the “reporting” and “non-reporting” surveillance sites was asked to complete a paper-based structured questionnaire between 15 January and 15 February 2020.

### 2.4. Data Variables, Sources of Data, and Data Collection

#### 2.4.1. Variables

The following variables were extracted from the Excel spreadsheet maintained at the NPHL to check the consistency, completeness, and timeliness of data in accordance with the GLASS reporting criteria: unique ID, type of specimen (blood, stool, urine, or genital swab), bacterial isolates (*Escherichia coli*, *Klebsiella pneumoniae*, *Salmonella* spp., *Shigella* spp., *Acinetobacter* spp., *Streptococcus pneumoniae*, *Staphylococcus aureus*, or *Neisseria gonorrhoeae*), antibacterial agents, age, sex, date of specimen collection, origin of the specimen, and detailed antibiotic susceptibility report. The overall consistency was assessed for this study by checking the variables used to generate the report (as directed by the GLASS manual i.e., four priority specimens and eight priority pathogens) while completeness signified no missing variables required according to the GLASS criteria: age, sex, pathogen, origin, specimen, antibiotic susceptibility results, and date. The dates that the monthly results were received at the NPHL were used to report the timeliness.

We used a structured questionnaire to collect variables, including human resource availability, AMR/WHONET software training, data analysis training, number of rooms available in the laboratory, dedicated space for data entry, computer and Internet service availability, agreement with the NPHL, availability of terms of reference, and institutional restrictions. As noted, one microbiology laboratory staff from each of the “reporting” and “non-reporting” sites completed the questionnaire.

#### 2.4.2. Sources of Data

The sources of data assessed were the monthly AMR data reports received at the NPHL (formats: Excel document, email, paper-based, or jpg file) and on-site AMR surveillance data from the “reporting” sites (registers/electronic database/software/WHONET file) only as the “non-reporting” sites had not sent the results for ninety consecutive days and hence the data could not be included in this short six-month study.

Responses to the structured questionnaire were the source of data regarding basic infrastructure and specific requirements for “reporting” and “non-reporting” sites (records and observation of basic infrastructure at the institution).

#### 2.4.3. Data Collection

Monthly AMR data reports received from 1 January to 30 June 2019, were assessed for completeness and consistency. On-site AMR surveillance data from the “reporting” sites were checked during visits in January–February 2020. “Non-reporting” sites were also visited during January–February 2020, and the microbiology staff responsible for data compilation were asked to complete the paper-based structured questionnaire. As noted, microbiology staff responsible for data compilation at the “reporting” sites also completed the questionnaire to allow for comparison. The principal investigator also observed the facilities after receiving the questionnaires.

### 2.5. Data Analysis and Statistics

All positive bacterial culture records from the Excel sheet at the NPHL were checked for consistency with the GLASS criteria (Appendix A) regarding recording the correct pathogen for the specific specimen type and the specific relevant antibacterial. The specimen–pathogen combination was defined as each priority specimen and its combination of priority pathogen e.g., genital swab and *Neisseria gonorrhoeae* or stool specimen and *Salmonella/Shigella* spp., and so on. The pathogen–antibacterial combination was defined as each priority pathogen and listed antibiotics to be tested e.g., *Staphylococcus aureus* with cefoxitin and oxacillin. The number and proportion of positive bacterial culture records consistent with the GLASS recommendations were recorded. While checking for completeness of pathogen–antimicrobial combination, the numbers included as total were adjusted according to the antimicrobial sensitivity for each pathogen. Colistin, though mentioned for Gram-negative organisms, was only included in the calculation if second-line antibiotics (e.g., carbapenems) were resistant and carbapenems were included in the calculation only if first-line antibiotics (e.g., third- and fourth-generation cephalosporins) were resistant. The data collected in the Excel sheet at the NPHL were verified with on-site visits to the five “reporting” sites. Before data analysis, the records were checked for duplicates and the ones occurring within a month (repeated isolates of the same bacterial species isolated from a patient within thirty days, regardless of specimen type) were removed. Timeliness was assessed as delay in days. Consistency, completeness, and timeliness were reported separately and cumulatively for the five sites.

The denominator for each site was its total number of checked positive records. The proportion was calculated using the following formula:(Number of consistent records/Total number of checked records) × 100

The Excel sheet at the NPHL was also verified with records at five reporting surveillance sites (paper-based at four sites and an electronic database at one site where the paper-based record was not available). The paper-based or electronic data at the surveillance sites was taken as the trusted source of data (“gold standard”).

If no discrepancy was found for any variable, a score of one was given for both consistency and completeness. If a discrepancy was found, a score of zero was given for consistency and a score of one for completeness. If there were any missing values in either the Excel sheet from NPHL or the source data, a score of zero was given for both consistency and completeness.

The specific number of variables for each pathogen in the Excel sheet was checked for completeness and consistency by comparison with source data fields. The proportion of data consistency was calculated by adding up the number of consistent data and dividing that value by the total number of expected data elements.

The timeliness was assessed by the timing of reports submitted to the NPHL. If a report was submitted by the 15th working day of the following month, a score of one was given. If it was submitted after the 15th working day, a score of zero was given. The number of days of delay in submitting reports was also calculated between the date of the deadline for each month and the date the report was submitted.

Differences in delays to report submission were assessed as the mean for the study duration, i.e., 6 months. Overall completeness and overall consistency among the five surveillance sites were also assessed. Box 1 explains the key operational definitions that were used in this study. The names of the surveillance sites were anonymized before any data analysis or publication of the results.

### 2.6. Ethical Approval

Ethics approval was obtained from the Union Ethics Advisory Group (International Union against Tuberculosis and Lung Disease, Paris, France) EAG No. 64/19 on 21 August 2019, and from the NHRC, Protocol No. ERB 673/2019P, on 29 December 2019. Approval was sought from all relevant stakeholders, including written permission for using the AMR surveillance data from the director of the NPHL and the AMR surveillance site institution/laboratory heads.

Box 1Operational definitions. ^1^ AMR, antimicrobial resistance; ^2^ GLASS, Global Antimicrobial Resistance Surveillance System; ^3^ NPHL, National Public Health Laboratory.AMR ^1^ dataDetailed identification and antibiotic susceptibility data of specific bacterial isolates, along with unique identifiers, specimen, origin, date of sampling, and demographic data from surveillance sites’ microbiology laboratory records.OriginPlace: “Hospital” or “Community” origin.Timeliness of dataAMR data, for a particular month, received within the 15th working day of the following month.Duplicate dataAMR data occurring within a month i.e., repeated isolates of the same bacterial species isolated from a patient within thirty days, regardless of specimen type.Specimen–pathogen combinationCombination of priority specimens (namely, blood, urine, stool, or genital swabs) with priority pathogens according to the GLASS^2^.Pathogen–antibacterial combinationCombination of eight priority pathogens and the relevant listed antibiotics according to the GLASS.Consistency of dataData is considered consistent when the variables used to generate the report are as directed by the GLASS manual i.e., four priority specimens and eight priority pathogens.Completeness of dataCompleteness signifies no missing variables required according to GLASS criteria: age, sex, pathogen, origin, specimen, antibiotic susceptibility results, and date.Non-reporting sitesThe surveillance sites that have not sent any AMR laboratory data to the NPHL^3^ for ninety days consecutively.Basic infrastructureBasic facilities and equipment required by the AMR surveillance site to send the AMR reports to the NPHL.Specific requirementsRequirements other than the basic infrastructure to send the AMR reports regularly to the NPHL.

## 3. Results

### 3.1. Consistency, Completeness, and Timeliness of AMR Surveillance Data

A total of 1584 records from five reporting sites were screened for consistency in the specimen–pathogen combination as per the GLASS criteria, with 1147 (72.4%) isolates assessed for urine, 403 (25.4%) for blood, 27 (1.7%) for stool, and 4 (0.3%) for genital swabs. Overall, 1571 (99.1%) records from all sites were found to be 100% consistent for blood, 96–99% for urine, 88.9–90.5% for stool, and 75% for genital swabs (Figure 1). In the figure below, the *x*-axis represents the five “reporting” AMR surveillance sites, and the *y*-axis denotes percentage consistency of specimen–pathogen combination for the four specimens indicated with different colored bars.

A total of 1584 records were assessed for the pathogen–antibacterial combination, with a 66% overall consistency. The consistency in reporting the pathogen and antibacterial sensitivity test according to GLASS criteria varied from 52% to 88% across the different sites, and the numbers varied among the organisms, too. The consistency was high for *Salmonella* spp. (94–100%), but the averages ranged from 54% to 83% for other bacteria such as *Staphylococcus aureus* (83%; range: 0–100%), *Shigella* spp. (71%; range: 0–86%), *Escherichia coli* (63%; range: 49–91%), *Acinetobacter* spp. (55%; range: 0–100%), and *Klebsiella pneumoniae* (54%; range: 48–70%). No consistency was found at any site for *Streptococcus pneumoniae* and *Neisseria gonorrhoeae* (Table 1).

When the AMR surveillance records at the sites were compared to records sent to the NPHL, there was completeness overall, except for a variable origin (hospital or community), which were missing data at two of the sites (sites B and C) and incomplete at two more (94% at Site A and 53% at Site D). For Site B, the antibiotic susceptibility, specimen–pathogen combination, and pathogen–antibacterial combination data were all incomplete (99.7%, 96.9%, and 97.9%, respectively) (Table 2).

Upon site visits, the specimen and isolate data were found to be 100% consistent at sites B and C, whereas the consistency at the remaining three sites varied from 84.8% to 99.1% for specimens and 84.8% to 94.5% for isolates. Only Site C had consistent data on the antibiotic susceptibility pattern; at the remaining four sites, the consistency varied from 80% to 94.5%. The specimen–pathogen combination and the pathogen–antibacterial combination consistency varied among the different sites, with ranges of 46.2–89.4% and 47.1–89.4%, respectively (Table 3).

The completeness of records from the “reporting” sites received at the NPHL varied from 88% to 100%, whereas the consistency varied from 77% to 92%. Data from only two sites were received by the 15th working day of the following month, whereas receipt was delayed for the other three by an average of 175 days (range: 8–269) (Table 4).

### 3.2. Barriers in Reporting AMR Surveillance Data

The structured questionnaire completed by the microbiology staff at the nine AMR surveillance sites indicated adequate staffing in the microbiology department. A dedicated room for data entry was available at only 50% of the “non-reporting” sites (Sites 1 and 2) whereas it was available at 80% of the “reporting” sites. The data entry area was in a shared space at Sites 4 and B. A computer for data entry with an adequate Internet connection speed was available at only two “non-reporting” sites (50%) as compared to all five (100%) “reporting” sites.

None of the sites had dedicated personnel available for data entry, which was being temporarily performed by laboratory personnel with a qualification ranging from certificate in Medical Laboratory Technology (CMLT), Bachelor of Science in Medical Laboratory Technology (BSc.MLT) to MSc. Microbiology.

Although most microbiology staff at all sites had received the AMR surveillance training, the data entry person at one of the “non-reporting” sites had not received the data analysis and WHONET training.

All of the sites had a Terms of Reference based on the 2013 AMR guidelines (published by the NPHL) and a verbal agreement with the NPHL to share the AMR data regularly. Three of the four “non-reporting” sites did not have any institutional restrictions on data sharing (Table 5).

## 4. Discussion

In this study, we assessed the quality of AMR data sent to the NPHL in Nepal from five of nine investigated AMR surveillance sites. Although considerable in volume, the data lacked completeness and consistency, with the main issues being related to the origin of the isolates (hospital or community setting). The data received at the NPHL also did not follow the GLASS criteria for the specimen–pathogen and pathogen–antibacterial combinations and often were submitted with delay. We looked as well at infrastructure availability at the surveillance sites, and among the four “non-reporting” sites, most lacked dedicated personnel for data entry and had not met some specific requirements, such as having a computer. Other researchers have stressed the need for infrastructure and human resources for reliable data in surveillance [11].

To track AMR across the globe, the WHO has created the GLASS with the aim of enabling international comparison, analysis, and sharing of AMR data, using information technology. Generation of quality AMR data requires good infrastructure; otherwise, it will be faced with challenges, particularly in resource-limited laboratory settings [12].

With our study, we have found a lack of uniformity between the stored data at the surveillance sites and the data received by the NPHL. Other researchers have also highlighted a non-uniformity of data that are kept in registers rather than computers [4,13]. Schnall et al. identified errors even in the electronic reporting of AMR surveillance data, caused by staffing constraints, which is also a key factor in the lack of reporting uniformity [12].

In Nepal, Malla et al. [14] had, by 2014, already highlighted the challenges to AMR surveillance implementation, giving high importance to data collection, compilation, and storage. Many others later noted that the availability of an electronic data collection system with data back-up, as opposed to manual paperwork collection, would result in a smoother workflow and better data flow from the surveillance sites to a central reporting site [15,16,17,18]. The use of WHONET software, with its uniform and standardized format, would facilitate an easy compilation for GLASS reporting [19].

Ethiopia has faced similar issues and pledged to develop a better AMR surveillance strategy. However, WHONET software integration for surveillance proved challenging at the data entry site, and they opted for an individualized solution for data entry [20].

Ibrahim et al. also found data transmission challenges in Ethiopia’s reporting to the GLASS, although they could improve it by on-site mentorship and frequent site visits with weekly calls [21]. In our study, the AMR surveillance site personnel had received training in how to enter data into WHONET, but two of the four “non-reporting” sites did not use the system because no computer was available.

At most of the surveillance sites, heads of the institution had not signed documents, such as an official memorandum of understanding or a written “Terms of Reference,” recording their commitment to report to the NPHL for the GLASS.

This study is among the first to assess the quality of data received for GLASS reporting, although others have highlighted the advantages of participating in the GLASS [17].

A limitation of this study is that it was conducted in only one province because of a lack of time and funding. These findings thus likely cannot be generalized to the whole country, given that only half of the surveillance sites could be assessed and only for a short period of time. For instance, one of the two “non-reporting” sites lacking a computer was in the process of receiving one.

Schnall et al. stressed the need for a consistent source of funding for a better surveillance system [12]. We wish to highlight that such monitoring and evaluation studies could be extensively planned if funding could be arranged and allocated. The way forward would be to expand this study to all of the surveillance sites in the country.

## 5. Conclusions

Our results provide a snapshot of the situation for GLASS reporting in 9 out of 21 surveillance sites in the country of Nepal. The completeness, consistency, and timeliness of the shared AMR data need improvement. Lack of dedicated data personnel and basic information technology infrastructure are important barriers to sharing data with the NPHL.

## Figures and Tables

**Figure 1 tropicalmed-06-00060-f001:**
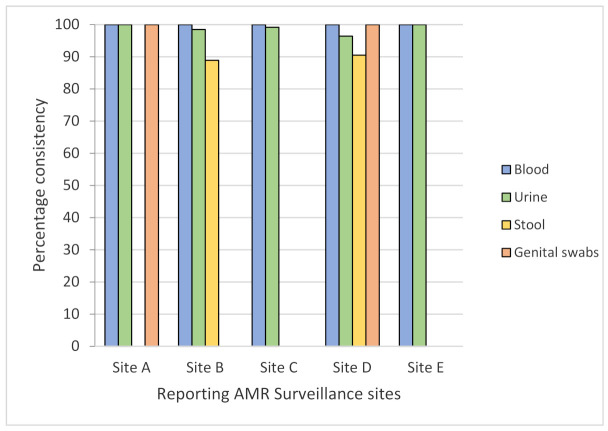
Consistency of specimen–pathogen combination according to the GLASS from five AMR surveillance sites in Province 3 of Nepal (January to June 2019).

**Table 1 tropicalmed-06-00060-t001:** Consistency in pathogen–antibacterial combination of AMR ^1^ surveillance data according to the GLASS ^2^ from five AMR surveillance sites in Province 3 of Nepal (January–June 2019).

Variables	Overall			Site A			Site B			Site C			Site D			Site E		
	Total (n)	Consistent (n ^3^)	%	Total (n)	Consistent (n)	%	Total (n)	Consistent (n)	%	Total (n)	Consistent (n)	%	Total (n)	Consistent (n)	%	Total (n)	Consistent (n)	%
Total number of records	1584	1038	66	428	223	52	381	221	58	372	296	80	232	204	88	171	94	55
*Escherichia coli*	1020	641	63	354	172	49	220	110	50	218	199	91	109	93	85	119	67	56
*Klebsiella pneumoniae*	242	133	55	31	15	48	67	37	55	94	50	53	10	7	70	40	24	60
*Acinetobacter spp. ^4^*	64	35	55	4	4	100	15	0	0	44	31	71	1	0	0	0	N/A ^5^	N/A
*Staphylococcus aureus*	77	64	83	4	0	0	19	19	100	16	16	10	29	29	100	9	0	0
*Streptococcus pneumoniae*	1	0	0	0	N/A	N/A	0	NA	N/A	0	NA	N/A	1	0	0	0	N/A	N/A
*Salmonella spp.*	159	153	96	34	32	94	56	55	98	0	N/A	N/A	66	63	95	3	3	100
*Shigella spp.*	17	12	71	0	N/A	N/A	3	0	0	0	N/A	N/A	14	12	86	0	N/A	N/A
*Neisseria gonorrhoeae*	4	0	0	1	0	0	1	0	0	0	N/A	N/A	2	0	0	0	N/A	N/A

^1^ AMR, antimicrobial resistance; ^2^ GLASS, Global Antimicrobial Resistance Surveillance System; ^3^ n, number; ^4^ spp., species; ^5^ N/A, not applicable.

**Table 2 tropicalmed-06-00060-t002:** Completeness of AMR surveillance data from five AMR surveillance sites in Province 3 of Nepal (January 2019–June 2019).

Variables	Site A		Site B		Site C		Site D		Site E	
	n ^1^	(%)	N	(%)	n	(%)	n	(%)	n	(%)
Total records	580	100	3164	100	810	100	265	100	341	100
Age	580	100	3164	100	810	100	264	99.6	341	100
Sex	580	100	3164	100	810	100	264	99.6	341	100
Origin	536	92.4	MD ^2^	0	MD	0	122	46	341	100
Date of isolation	580	100	3164	100	810	100	232	87.5	341	100
Specimen	575	99.1	3164	100	810	100	232	87.5	289	84.8
Isolate	548	94.5	3164	100	810	100	232	87.5	289	84.8
Antibiotic susceptibility results	548	94.5	3126	98.8	810	100	232	87.5	273	80
Specimen–pathogen combination	428	73.8	1461	46.2	724	89.4	232	87.5	289	84.8
Pathogen–antibacterial combination	428	73.8	1490	47.1	724	89.4	232	87.9	289	84.8

^1^ n, number; ^2^ MD, missing data.

**Table 3 tropicalmed-06-00060-t003:** Consistency of AMR surveillance data from five AMR surveillance sites in Province 3 of Nepal (January to June 2019).

Variables	Site A		Site B		Site C		Site D		Site E	
	n ^1^	(%)	N	(%)	n	(%)	n	(%)	n	(%)
Total records	580	100	3164	100	810	100	265	100	341	100
Age	580	100	3164	100	810	100	264	99.6	341	100
Sex	580	100	3164	100	810	100	264	99.6	341	100
Origin	536	92.4	MD ^2^	0	MD	0	122	46	341	100
Date of isolation	580	100	3164	100	810	100	232	87.5	341	100
Specimen	575	99.1	3164	100	810	100	232	87.5	289	84.8
Isolate	548	94.5	3164	100	810	100	232	87.5	289	84.8
Antibiotic susceptibility results	548	94.5	3126	98.8	810	100	232	87.5	273	80
Specimen–pathogen combination	428	73.8	1461	46.2	724	89.4	232	87.5	289	84.8
Pathogen–antibacterial combination	428	73.8	1490	47.1	724	89.4	232	87.9	289	84.8

^1^ n, number; ^2^ MD, missing data.

**Table 4 tropicalmed-06-00060-t004:** Difference in completeness, consistency, and timeliness (with report submission delay) from five AMR ^1^ surveillance sites in Province 3 of Nepal (January to June 2019).

Variable	Site A		Site B		Site C		Site D		Site E	
	n ^2^	(%)	n	(%)	n	(%)	n	(%)	n	(%)
Completeness	3826	99	3031	88	2976	89	1978	95	1539	100
Consistency	4803	92	21897	77	6308	87	2042	86	2793	91
Timeliness	0	-	0	-	1	-	0	-	1	-
Mean delay (days)	8	-	247	-	0	-	269	-	0	-

^1^ AMR, antimicrobial resistance; ^2^ n, number.

**Table 5 tropicalmed-06-00060-t005:** Assessing the availability of basic infrastructure and specific requirements in four “non-reporting” AMR ^1^ surveillance sites in Province 3 of Nepal (January to February 2020).

Requirements	Response			
	Site 1	Site 2	Site 3	Site 4
Number of microbiology staff	15	8	4	7
Number of rooms dedicated to data entry	One	One	None	None
Area of data entry room	<150	<150	N/A ^2^	<150
Availability of computer for data entry	Yes	No	No	Yes
Number of computers for data entry	5	One	N/A	1
Availability of Internet service	Yes	No	N/A	Yes
Speed of Internet service	>0.5 Mbps ^3^	N/A	N/A	>0.5 Mbps
Availability of person for data entry	Yes	No	No	Yes
Qualification of data entry person	BSc.MLT ^4^	BSc.MLT/CMLT ^5^	N/A	BSc.MLT
Training received on AMR surveillance	Yes	Yes	Yes	No
When was AMR surveillance training received?	2019	2019	2019	N/A
Training received on data entry and analysis	Yes	Yes	Yes	No
When was data entry training received?	Every year	May 2019	May 2019	N/A
Training received on WHONET ^6^	Yes	Yes	Yes	No
When was WHONET training received?	2019	2019	2019	N/A
Agreement/TOR ^7^	Verbal	Verbal	Verbal	Verbal
Institutional restrictions on data sharing with the NPHL ^8^	None	None	Verbal	None

^1^ AMR, antimicrobial resistance; ^2^ N/A, not applicable; ^3^ Mbps, megabytes per second; ^4^ BSc.MLT, Bachelor of Science in Medical Laboratory Technology; ^5^ CMLT, Certificate in Medical Laboratory Technology; ^6^ WHONET, WHO free software; ^7^ TOR, terms of reference; ^8^ NPHL, National Public Health Laboratory.

## Data Availability

The data contains sensitive information that was obtained from various AMR surveillance sites after approval from the relevant authorities and in-country ethics committee. We have permission to share only aggregate, analyzed data but not individual patient-wise data. Therefore, the data cannot be made available publicly. However, if anyone gives justifiable reason to access the individual patient-wise de-identified data, they are requested to contact the corresponding author (jyotigan30@gmail.com) or National Public Health of Nepal Email: nphl@nphl.gov.np (institutional email address). Mailing address: National Public Health Laboratory, Department of Health Services, Ministry of Health and Population, Tripura Marg, Teku, Kathmandu, Nepal. Postal code 44600. Tel.: +977-9841292279; +977-01-5352421.

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
