# Peer review of "Quality Assessment of an Antimicrobial Resistance Surveillance System in a Province of Nepal"

_tropicalmed, 2021, doi:10.3390/tropicalmed6020060_

Round 1

Reviewer 1 Report

Antibiotic resistance is a major threat to human health. The WHO has set up a surveillance program for antibiotic resistance according to a standardized protocol that allows to compare the evolution of the problem over time and between countries. This manuscript describes the implementation of this program in nine laboratories from a representative province of Nepal. The objectives of the study were to analyse both the quality of the data collected and the structural functioning of the local laboratories in charge of the surveillance. This approach points out the problems potentially encountered during the collection and analysis of resistance data. It also points some organizational issue at the laboratory level.

A few points need to be clarified to help recontextualize the study:

1) The representativeness of the laboratories included in the study could be more detailed. Are these laboratories attached to hospitals or rather clinics of outpatients? How many samples per year are analysed in each lab? These two points may explain the availability of certain data and the experience of operators, the resource made available. Hospital labs will not have the same type of samples that of labs in charge of outpatients. They will test different antibiotics more adapted to their type of population.

2) The authors should clarify items showing low completeness rate depending on the pathogens included. Some labs might not test routinely certain antibiotics. Some of them might not be available in Nepal. In other cases, it could be antibiotics of limited utility for the study population (e.g. meropenem for E. coli in urine from outpatients). This useful information could subsequently guide public health authorities in making recommendations for antimicrobial susceptibility testing. Other factors could explain problems with the completeness of antibiotic susceptibility data. For example, some antibiotics are difficult to test with the methods routinely used by laboratories (e.g. colistin is very method dependent. The disk diffusion method which is used by many labs around the world do not allow testing for colistin sensitivity)

3) Did the authors check the quality of the deduplication which is essential for correct resistance data. Did they check the ability/knowledge of the labs to remove duplicates according to the GLASS protocol recommendations?

4) Figure 1 should be explained with a complete legend. What do the diagrams correspond to?

5) The authors should define the following terms “specimen pathogen combination” and “pathogen antibacterial combination”. Depending on the definition this could explain the low rates of consistency or completeness.

6) In order to compare sensitivity/resistance results, laboratories must use the same AST interpretative criteria. GLASS recommends using either the American CLSI or the European EUCAST standards. Which criteria are used in Nepal?

Reviewer 2 Report

None

Author Response

Thank you for devoting your precious time to review the manusript.

Reviewer 3 Report

The study aimed to assess and validate AMR surveillance in one province of Nepal. The study is well designed and the methodology is sound. The introduction is well written and provides context to the international audience. Surveillance data are assessed as well as an exploration of  barriers that laboratories may face in supplying data. The study is a mixed methods - quantitative and qualitative data which is appropriate. The methodology section did not supply sufficient detail to judge the  quality of AMR data that were being supplied to the NPHL, it was not clear from the description provided whether the data were deduplicated and / cleaned before supplied to the national laboratory. This is an important component of data validity which is not clear in the methods section.  The methods section also lacked clear description of the survey design, delivery and analysis (i.e thematic?). As the aim of the study was to assess validity of the amr data, at this stage I am not certain how this was done (or what gold standard it was assessed against). I can appreciate that there aren’t many data to compare their surveillance data against but some measure of validating the quality of AMR data would be needed to ensure that what is sent to GLASS is appropriate.

Specific comments

  • Line 129 : what is ‘positive laboratory record’ referring to? Positive culture? or something else?
  • What are the GLASS criteria for ‘consistency’ ‘ completeness’ and ‘timely’, or are these the ones in box 1. If so, a ref to GLASS should be made
  • Why was the data from ‘non-reporting’ sites not collected for the assessment (only collected for survey)? this would be a good comparison for data quality and completeness against those sites who do already report.
  • How were data cleaned and processed prior to reporting to the NPHL. In particular I would like to see comment on deduplication of data (i.e. time frames between isolates);
  • Structured questionnaire should be included (can be in supplementary) as its difficult to judge based on the information provided whether the survey questions were appropriately designed and measured the intended purpose.
  • How have the authors ‘validated’ the AMR data ? this was an aim in the study but I am unsure how this was carried out.
  • Figure 1 – unsure of what each coloured column represents. Need a key or further detail
  • Barriers in reporting AMR surveillance, line 258 – this paragraph needs further explanation? Where are the data for the ‘reporting ‘sites? how many people were asked to participate in this survey, how many completed the survey (i.e. response rate), was it an online anonymous survey or paper-based or otherwise? Unsure what the purpose of asking the ‘area of data entry room’ is this referring to square m?

Round 2

Reviewer 3 Report

nothing further to add